# Robustness Evaluation Process for Scheduling under Uncertainties

**Sara Himmiche** [1,*]**, Pascale Marangé** [2]**, Alexis Aubry** [2] **and Jean-François Pétin** [2]

1    ICube Laboratory, University of Strasbourg, 67000 Strasbourg, France
2    CRAN, CRNS, University of Lorraine, F-54000 Nancy, France
*    Correspondence: himmiche@unistra.fr

**Abstract:** Scheduling production is an important decision issue in the manufacturing domain. With the advent of the era of Industry 4.0, the basic generation of schedules becomes no longer sufficient to face the new constraints of flexibility and agility that characterize the new architecture of production systems. In this context, schedules must take into account an increasingly disrupted environment while maintaining a good performance level. This paper contributes to the identified field of smart manufacturing scheduling by proposing a complete process for assessing the robustness of schedule solutions: i.e., its ability to resist to uncertainties. This process focuses on helping the decision maker in choosing the best scheduling strategy to be implemented. It aims at considering the impact of uncertainties on the robustness performance of predictive schedules. Moreover, it is assumed that data upcoming from connected workshops are available, such that uncertainties can be identified and modelled by stochastic variables This process is supported by stochastic timed automata for modelling these uncertainties. The proposed approach is thus based on Stochastic Discrete Event Systems models and model checking techniques defining a highly reusable and modular process. The solution process is illustrated on an academic example and its performance (generecity and scalability) are deeply evaluated using statistical analysis. The proposed application of the evaluation process is based on the technological opportunities offered by the Industry 4.0.

**Keywords:** robustness evaluation; production scheduling; uncertainties; discrete event systems; decision making; Industry 4.0





## 1. Introduction

The expectations of industries are increasing: to manufacture in shorter and shorter time frames, with higher and higher quality and with the possibility to customize any product on demand. In this context, industries seek to define optimal production schedules, i.e., to find the solution that defines the start and completion dates of tasks, and the allocation to dedicated resources. Approaches to scheduling, which are generally sought to be optimal in terms of product set, production time or resource utilization, are the subject of a large literature, particularly from operations research. These different approaches are generally carried out in a predictive manner by considering a stable environment in terms of demand and resources. Historically, approaches to optimal scheduling have been extensively discussed in the literature, especially in the operational research (OR) community [1]. This strong assumption on the stability of the information used to calculate the scheduling solution is difficult to maintain. In order to take these uncertainties into account, there are two possibilities: either to recalculate the scheduling online [2], or to take the uncertainties into account when calculating the scheduling and evaluate different indicators. The work proposed in this paper falls into the second category. In fact, optimal scheduling risks seeing its performance deteriorate during its implementation [3], given the inevitable deviations from the environment in practice: uncertainties and high variability on the sets of products to be produced (volume, mass

customization, production of small series, increasingly short lead times between order taking and delivery time, etc.), uncertainties on the production resources (operating times, machine breakdowns, manufacturing hazards, etc.) Manufacturers currently need other indicators to make their scheduling choices, for example, delay times in the event of a breakdown, the absorption capacity of their scheduling in the face of uncertainties about production times. The work proposed in this paper therefore focuses on the evaluation of a scheduling system in the face of uncertainties and thus proposes different indicators to help the decision-maker choose a solution.

Production scheduling under uncertainty is not a new issue. The main propositions to tackle this problem focus on generating robust schedules that guarantees a performance level [4]. These methods are specific to the production problem treated. The re-usability of proposed approaches to other problems is then difficult.

In this paper, we propose a generic evaluation approach that can, on the one hand, be adapted to different types of workshop and different types of uncertainty and, on the other hand, evaluate the robustness of an initial production schedule. For this purpose, we use discrete event system modelling and statistical model-checking based on Monte Carlo simulation. A numerical example for illustrating the application of the of the process is given and a deep discussion, regarding the process performances from the genericity, sensitivity and scalability point of views, is provided.

## 2. Literature Review

Even if scheduling under uncertainties is still a critical issue, it is not a new research issue. More generally, decision making and optimization under uncertainties are topics that have been extensively studied in the last decades. Twenty years ago, Ref. [5] already justified the need of hedging uncertainties in optimization by the fact that for real-world optimization problems, the "decision environment" is often characterized by: (i) uncertain/inexact data, (ii) the difficulty to implement accurately an optimal solution (even if computed very accurately), (iii) the necessity to satisfy the constraints for all meaningful realizations of the data, (iv) the fact that "Bad" optimal solutions (those which become severely infeasible in the face of even relatively small changes in the nominal data) are not uncommon. Thus, to not consider these uncertainties in optimization in general and thus in scheduling in particular can lead to unworkable solutions. Regarding the scheduling problem, Ref. [6] advanced that *the inability of much scheduling research to address the general issue of uncertainty is often cited as a major reason for the lack of influence of scheduling research on industrial practice*. Ref. [7] went a step further and advanced that *optimization is actually the opposite of robustness*.

This ability to address uncertain knowledge in scheduling remains a need that is clearly identified as a key performance in the recent literature. Ref. [8] proposed a literature review concerning smart production planning and control. Based on this literature review, they identify required smart capabilities such as *smart shop floor control and scheduling capability*, and for reaching these capabilities, they determine key performances. *Improve manufacturing robustness* is one of them. Ref. [9] proposed another review that is more dedicated to production scheduling in the context of Industry 4.0. The authors identified *scheduling under uncertainty, incomplete and missing data* as one of the critical scheduling areas. Moreover, they pointed the fact that the complexity of the problems to be solved will increase, as a consequence of the complexity of the real-world situations. More recent papers are aligned with these conclusions. Ref. [10] talks about *disturbance- and disruption- resistant scheduling*. Ref. [11] analysed existing Job Shop Scheduling contributions in the context of Industry 4.0 and concluded that they are not sufficiently *focused on the emerging trend of robustness and smart scheduling*.

According to [1] a Manufacturing Scheduling System is defined as in Figure 1. It consists in three main types of modules:

- database, object base, and knowledge-base modules,
- modules that generate the initial schedules, and

- user interface modules.

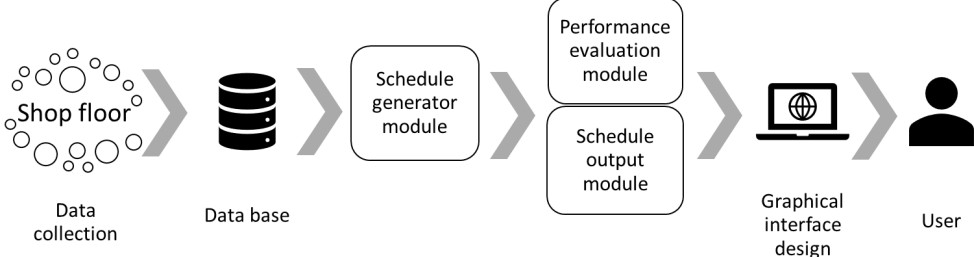

**Figure 1.** A scheduling system (figure modified from [1]).

Clearly, the user interface modules are important for implementation success. Their goal is to give to the decision maker the opportunity to modify the scheduling solution manually and to receive performance indicators for these solutions for taking the right decision. This is particularly true in the context of Industry 4.0.

Robustness is one of these indicators when dealing with uncertainty.

Ref. [7] defined a solution as robust if *its performance remain relatively unchanged when exposed to uncertain conditions*. This definition is typically consensual but needs to preliminary answer the two following questions:

1. How to qualify the uncertainties and how to integrate them into the scheduling problem?
2. How to measure the fact that a solution remains efficient despite these uncertainties?

### 2.1. Modelling Uncertainties

Regarding the first question, we would like to know what means uncertainties (theoretically, all the input data of the problem can be impacted) and then to propose a model for catching them. Ref. [12] proposed a typology for uncertainties: data can be uncertain (subjected to doubt concerning the validity of knowledge), incomplete (subjected to a lack of knowledge) or imprecise (subjected to a lack of precision). In literature, uncertainties are classified into several types. For instance, Ref. [13] categorize uncertainties into two groups linked to the source of uncertainty. It may come either from (i) environment or (ii) production system. In this paper, we combined the last classification with a second criteria linked with the impact of uncertainty (Table 1). In fact an uncertainty is characterized by both its source and its type. For instance, an urgent order comes from the environment of the production system and is linked to event occurrence.

**Table 1.** Uncertainty classification.

|  | **Uncertainty on Parameters** | **Uncertainty on Events** |
|---|---|---|
| **Environmental** | Demand volume<br>Supply delivery time<br>… | supply disruption<br>Urgent order<br>… |
| **System** | Operation duration<br>Reparation duration<br>… | Machine breakdown<br>Product defect<br>… |

Ref. [14] proposed another typology: Uncertainties on parameters (noted *UoP*) and Uncertainties on events (noted *UoE*). *UoP* are defined as the difference between the predicted information and the real available information whereas *UoE* are defined by the occurrence of uncontrollable events in the system or its environment. Modelling uncertainties is the underlying question. Ref. [15] present the different uncertainty models and the associated approaches that are used in decision aiding and optimization problems. They also propose a framework that describes the relationships among them. Ref. [16] is more focused on scheduling under uncertainties and listed the three most usual approaches

used to model uncertainties: the bounded form, the probability description, and the fuzzy description.

- The *bounded form* is used when it is possible to maintain values of data in bounded limits. Then, the uncertainties can be described by a set of scenarios which can be discrete or continuous inside these bounded limits.
- The *probability description* or stochastic modelling is used when there exist historical and statistical data. The uncertainties can then be modeled as random variables that follow a discrete or continuous probability distribution.
- The *fuzzy description* method is based on the possibility theory. When problem data are imprecise or not available, the uncertainty parameters are modeled in fuzzy sets and linked to satisfaction functions.

The probability description is relevant when information about the behavior of uncertainty is available. The bounded form is typically relevant when this information is not enough in order to develop an accurate description of the probability distribution, but only error bounds can be obtained [16]. The fuzzy description [17] is relevant when historical data are not readily available. Regarding the Industry 4.0 context, it is reasonable to consider that data coming back from the shopfloor can be used to build sufficiently rich information on the uncertainties. Thus, a probability description seems particularly relevant in this context.

### 2.2. Schedule Robustness as Performance Indicator

Regarding the second question identified in introduction, we would like to know how to measure the robustness of a solution such that this solution can be computed or chosen. Note that the answer to this question is deeply linked with the uncertainty model.

Robustness can be seen as a performance indicator as it measures the ability of a solution to maintain its performance relatively unchanged when exposed to uncertain conditions [7]. More specific KPIs when considering reliability only can be found in [18].

For considering general concerns in the robustness topic, Ref. [7] surveyed the different robustness approaches associated to the uncertainties models.

Considering the bounded form uncertainties, the common robust measure is the robust counterpart, also called minmax-robustness. This approach is detailed in [19] and consists in evaluating the performance in the worst case regarding the bounded set of uncertainties. The most common robustness metrics were presented in [20] and are still valid. Refs. [21,22] are surveys dedicated to the minmax-robustness applied to classical optimization problems such as the shortest path problem or the knapsack problem. Ref. [23] moreover discussed the different application of minmax-robustness as finance, revenue management, energy systems and scheduling.

The main drawback of the minmax-robustness approach concerns the fact that it is a conservative approach in the sense that it is *based on the anticipation that the worst might happen* [20].

Considering the uncertainties modeled by fuzzy description, the possibility theory provides the theoretical basis for defining relevant metrics based on the concepts of belief and plausibility [7]. For an application of fuzzy description in scheduling, the reader can refer to [24] or [17].

Considering the uncertainties modeled by probability description, different approaches can be discussed [7]: the expected value of a utility function, the probabilistic threshold (or service level) that evaluates the probability that the cost of a solution satisfies a given threshold, statistical feasibility robustness that consists in guaranteeing the constraints statistically (the constraints have a sufficient probability to be satisfied).

When focusing only on scheduling under uncertainty, Ref. [25] proposed a survey on single machine scheduling under uncertainties and Ref. [26] proposed a survey on flowshop scheduling under uncertainties. Refs. [27,28] focused on the parallel machine scheduling problem under uncertainty but proposed a state-of-the-art regarding this problem that can be generalized to scheduling under uncertainty. The literature review confirmed the

used of the three types of models and the associated robustness metrics. Second, they pointed that uncertainty in scheduling is receiving more and more attention. Moreover, Ref. [26] concluded that probability description is the most used model.

Regarding the integration of new enabling technologies in scheduling practices, big data and machine learning techniques have already been considered as emerging trends. Ref. [29] reviewed the recent progress of data-driven mathematical programming under uncertainty. Ref. [30] proposes an application of deep reinforcement learning for lexible job shop scheduling problem. Ref. [31] proposed an architecture for a scheduling systems based on data-driven procedures. Such types of works suggest that it will be possible in a next future to extract information directly from the shopfloor (The shopfloor data collection system). This will be true also for the vital information concerning the uncertainties such that the probability description model seems to be a convenient model in the context of Industry 4.0.

### 2.3. Discussion and Solution Direction

Regarding the previous state-of-the-art, one commonality relating to the reviewed works is that the solution approaches that are used are dedicated to one particular problem (single machine, parallel machines, flow-shop, . . . ) and to specific uncertainties.

Moreover, in this context, the question of performance evaluation is a key issue, and robustness must be considered as an important performance for dealing with the inherent uncertainties meaning that it must be considered in the "Performance evaluation" module.

The next step is thus to propose an evaluation process that is (i) able to evaluate the robustness of a candidate scheduling solution and, (ii) is generic regarding the type of the workshop and the considered uncertainties. Then the solution maker (the human in the loop) is able to take the best decision.

In order to achieve this goal we propose an evaluation approach that answers the following research questions:

- **RQ1** How to give, to the decision maker, a modelling framework for his/her schedules subjected to uncertainties that is: (i) generic, (ii) modular and (iii) reusable?
- **RQ2** How to give, to the decision maker, a tool for evaluating the robustness of his/her schedule modelled using the framework that answers to RQ1?

In the following section, an overview of the global process for answering these two questions is given. RQ1 is particularly addressed by proposing a modelling framework based on stochastic timed automata that is detailed in Section 4. RQ2 is particularly addressed by proposing a tool, based on simulation and model-checking techniques, that is detailed in Section 5.

Moreover, in the remaining of the paper, the following assumptions are made.

- **AS1:** uncertainties data are available in the workshop (Industry 4.0 assumption).
- **AS2:** uncertainties are described with stochastic models.
- **AS3:** a predictive schedule is known and has been generated before the evaluation.

### 3. Overview of the Process for Robustness Evaluation of Scheduling under Uncertainties

In the classical scheduling problem, the data of the workshop are stated as certain and static. In reality, the performance or even the admissibility of the predictive scheduling solution can be questioned when unpredictable events occurs or when real data are different from predicted ones. To ensure the admissibility and the performances of a predictive schedule to the real situation of the workshop, the problem of production scheduling must be upgraded to deal with the uncertainties.

### 3.1. Scheduling Problem Notations

To formulate the scheduling problem, three information sets are thus needed: the workshop, the constraints, and the objectives. The workshop information are defined by:

- A set $J$ of jobs $j$ to be processed in the workshop. With $NbJ$, the number of jobs;
- Each job $j$ is associated to a set $O_j^J$ of operations $o_{jk}$ defining a necessary routing to perform the job. With $NbOp^j$, the number of operations to realize the job $j$;
- A set $R$ of resources $r$, allowing the execution of operations. With $NbR$, the number of workshop resources;
- In order to execute the operation $o_{jk}$ on the resource $r$, an execution duration is defined as $d_{jkr}$.

The second information set is the workshop constraints that must be satisfied by a scheduling solution to be qualified as admissible. There are first the structural constraints that are related to the type of the workshop. In a Flow Shop and Job Shop, the operations of the same job have precedence constraints defined by the job route. Moreover, in a Flow Shop, all the jobs follow the same route. For the Open Shop, the operations in the job route can be executed in any order. The Hybrid Shop is a mix between these different types. Other constraints can be considered in the scheduling problem (preemption, batch production, release dates, etc.) [1].

A scheduling solution, denoted as $s$ can be represented as a Gantt diagram [32] (as in Figure 2) in which we can deduce:

- The allocation of resources to operations and the sequence of operations on each resource. For instance, in the solution of Figure 2, $r_1$ is allocated to $\{o_{11}; o_{22}; o_{33}\}$ and the sequence on this resource is $o_{11} \rightarrow o_{22} \rightarrow o_{33}$;
- The total predicted duration of the schedule $s$, denoted as $C_{max}^{ref}(s)$. For instance in Figure 2, $C_{max}^{ref}(s) = 18 TimeUnit$ .

Note that the solution of Figure 2 satisfies the job shop constraint as all the operations of the same job are processed sequentially according to the order of the route ($o_{21} \rightarrow o_{22} \rightarrow o_{23}$, etc.).

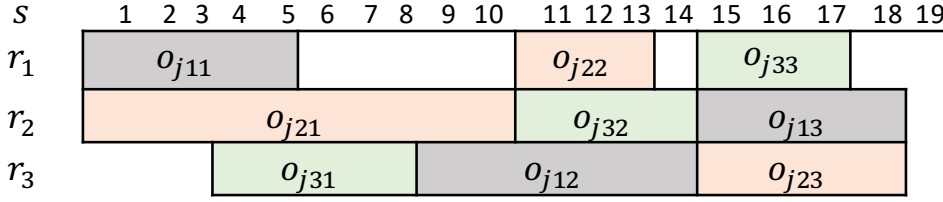

**Figure 2.** Schedule example.

*3.2. Robustness Evaluation Process*

The proposed process is the result of several reflections on the issue of robust production scheduling under uncertainties. Indeed, in [14], we have proposed a first approach that allows to deal specifically with the uncertainties on the execution durations of operations since it has a consequent impact on the makespan criterion of a schedule. While in [33] the uncertainties related to the machine failure and its reparation duration is analyzed in a specific approach. In the following, the evaluation process has evolved to a generic and adaptable one in order to stay independent of the type of workshop treated and the uncertainty to be considered.

The main objective of the evaluation process is to help the decision maker to choose the adapted "robust" schedule that satisfies workshop constraints and a desired deadline despite considered uncertainties. In the context of Industry 4.0, the process (Figure 3) is considered as a decision making process allowing the using of available data collected and treated in the workshop, as defined by [1]. The process can either be integrated in a scheduler as an evaluation module or applied by the decision maker on existing schedules.

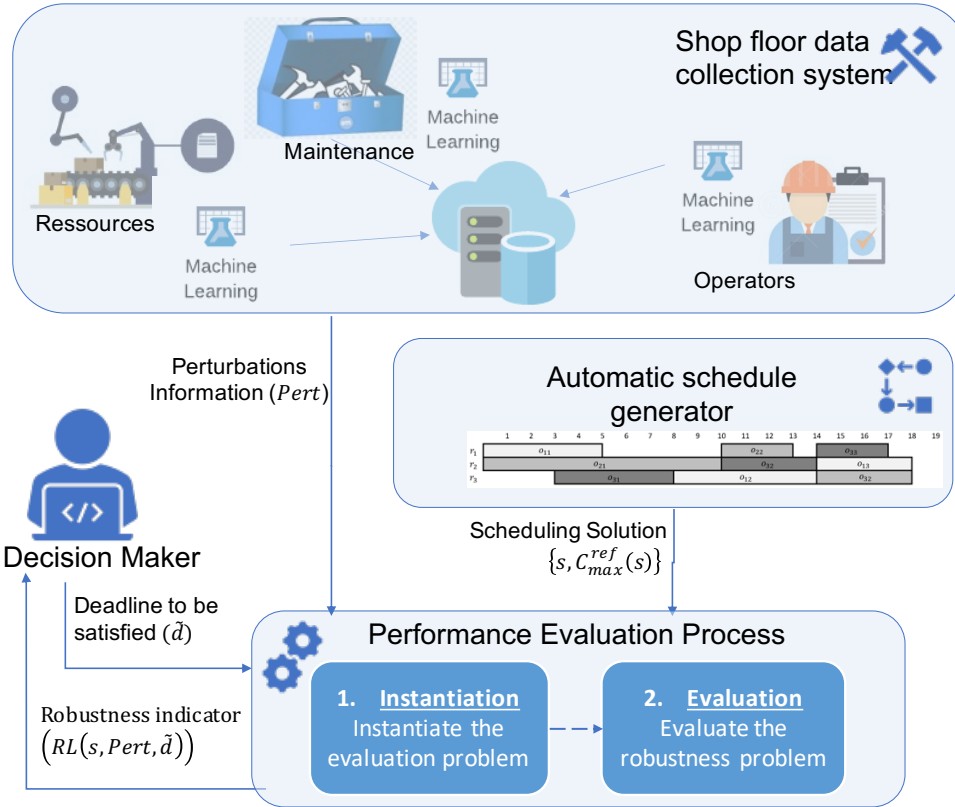

**Figure 3.** Robustness evaluation process.

The process has three inputs given by the decision maker or the schedule generator:

- Schedule $s$ is defined by the set of jobs and their related operations, the sequence of operations on resources and its initial makespan $C_{max}^{ref}(s)$.
- Set of uncertainties $U$ that have to be considered.
- Deadline $\tilde{d}$ that must be guaranteed despite the uncertainties.

The evaluation process is a two-step process. First, a set of generic models is instantiated according to the input data. Then, the robustness level is evaluated on the instantiated models and sent to the decision maker. To be implemented in an Industry 4.0 context, the evaluation process must satisfy the following challenges:

1. Genericity in front of the types of workshops(Job Shop, Flow Shop, Open Shop, etc.).
2. Genericity in front of the type of uncertainties considered, their stochastic data and the number of uncertainties considered.

The next section will present in detail the generic models that can be instantiated by the decision maker (step 1 in Figure 3) whereas Section 5 will present the evaluation step by using model-checking on the instantiated models.

## 4. STA Based Models for Scheduling under Uncertainties

### 4.1. Generic Modelling Approach

To model both schedules and uncertainties, a modular approach based on the concept of plug and play is suggested. In fact, our approach is based on several patterns that will model the behavior of the schedule elements and the uncertainties that will occur in the workshop. The main interest of such structure in the Industry 4.0 is to allow the adaptability of the process to the input data. The plug and play approach allow then to re-use the process with different uncertainties and different schedules.

On one hand, the schedule is modeled using two patterns. The first one presents the behavior of an operation that needs to be executed, while the second one models the behavior of a resource (machine, operator, etc.). On the other hand, uncertainties are

presented by two patterns, the first one models the behavior of uncertainty on parameters (UoP) and one the behavior of uncertainty on events (UoE).

The choice of having separate models for each element is to satisfy the need to evaluate the schedule performance regarding to different uncertainties and different workshops. Actually, considering a new uncertainty or increasing the number of uncertainties does not impact all models. Thus, it is contributing to the targeted genericity face to the type of the uncertainties. Moreover, the changing of workshop type will not affect the uncertainty models, contributing to the targeted genericity face to the type of workshop.

This concept generates several modelling constraints. The modelling tool to be used has to allow the modular modelling (to make it adaptable to the problem parameters), dynamic modelling (allowing the communication between patterns), stochastic modelling (to take into account stochastic models of uncertainties) and instantiable models (to be adaptable to the problem size).

Stochastic Timed Automata Formalism

To model the behavior of schedules and uncertainties, we should be able to satisfy the modelling constraints such as communicating models, time characteristics and probabilistic behavior of uncertainties.

Usually, the scheduling problem under uncertainties is modeled using Operational Research tools such as mathematical programming [20,34]. To meet the modelling constraints mentioned above another alternative is to use Discrete Event Systems (DES). In the field of safety and security, DES are generally used to evaluate reliability, availability and maintainability. Thus, they have proven their effectiveness in modelling stochastic and dynamic systems and evaluating system properties. Moreover, DES present considerable advantages making them good candidates for solving the scheduling problem in general [35]. Many stochastic DES languages allow the modelling of some of these characteristics: Stochastic Petri nets [36], Stochastic automata [37] and Stochastic automata networks [38]. The stochastic timed automata (STA) language answers to all modelling criteria and is therefore chosen. The STA language is an extension of the timed automata [37] which is enriched with shared variables, synchronizing events and probabilistic characteristics [39].

**Definition 1.** *Formally a Stochastic timed automaton is presented as the following n-tuple* $A = (L, V, E, C, Inv, Pr, T, L_m, l_0, v_0)$ *where:*

- *$L$ is a finite set of locations.*
- *$V$ is a finite set of variables.*
- *$E$ is a finite set of synchronizing events with $E = E_u \cup E_{\bar{u}}$.*
    - *$E_u$ is a finite set of urgent events. To prevent the network of automata from delaying when two components are able to synchronize, an event can be declared as urgent. In other words, transitions must be fired as soon as the guards are satisfied, without allowing time to pass.*
    - *$E_{\bar{u}}$ is a finite set of non-urgent events.*
- *$C$ is a finite set of clocks.*
- *$Inv$ is a set of invariants (conditions in location).*
- *$Pr$ is a set of probabilities: (i) discrete for the set of transitions (from a location $l_i$, probabilistic transitions allows to attend different locations $l_j$ with a given probability $p_{ij}$, with $p_i = \sum_{j=0}^{n} p_{ij} = 1$. (ii) continuous for the variables (the crossing condition of a transition is defined randomly by a probability distribution).*
- *$T$ is a finite set of transitions $(l, e, g, m, l') \in L \times E \times G \times M \times L$ where $l$ and $l'$ are respectively the starting and arriving locations . On a transition, three optional elements are defined: (i) a guard (condition on variables) $g$ from the set of guards $G$, (ii) an update (on variables) $m$ from the set of updates $M$, (iii) and a synchronizing event $e$ from the set $E$.*
- *$L_m \subseteq L$ is the set of marked locations.*

- $l_0 \in L$ *is the initial location of the automaton.*
- $v_0$ *is the initialization vector of variables.*

The elements of an STA are graphically represented as follows. Locations are represented by vertices, marked location by a double vertices, transitions by arcs and the initial location by an initial arc. An invariant is represented inside the associated vertex (location). Guards are represented between brackets "[ ]". Synchronizing events are denoted in italic. The update of variables and clocks are placed between parenthesis "()". Discrete probabilities are modeled by dotted arcs and associated probabilistic values are underlined. For continuous probabilities, they are directly linked with the definition of variables.

**Example 1.** *Let us consider two automata STA (Auto$_A$) and (Auto$_B$) (Figure 4) as a representative construction example of two automata synchronized with a synchronizing event. Three variables are defined: two clocks c for (Auto$_A$) and x for (Auto$_B$) and a synchronizing event $e_1$. The behavior of the two automata is constrained by the synchronizing event $e_1$ and the probabilistic behavior is modeled by two discrete probabilities.*

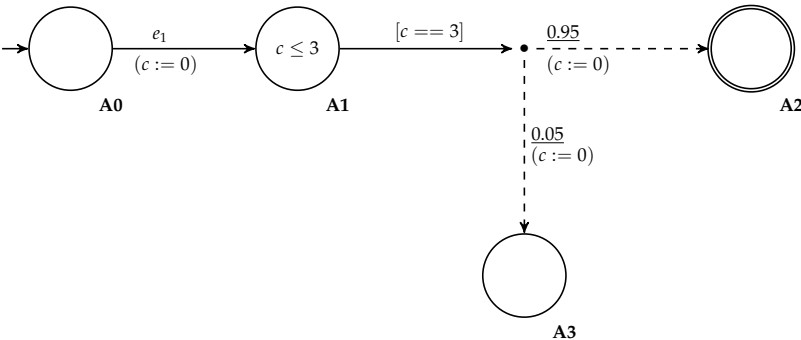

(**a**) Automate (*Auto$_A$*).

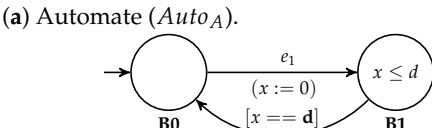

(**b**) Automaton (*Auto$_B$*).

**Figure 4.** Example of a stochastic timed automata.

In the following, the behaviors of schedule and uncertainties are modeled by several STA patterns. First, the UoP and UoE models are presented. Second, the operation model and resource model are explained. These models communicate using synchronizing events and shared global variables.

### 4.2. Modelling Uncertainties
#### 4.2.1. Impact of Uncertainties on Scheduling Solution

To take into account the different uncertainties in the initial predictive schedule, the following assumptions are made:

- uncertainties have an impact on the execution duration of operations, i.e., they generate a variation on the execution duration of operations and thus on the overall duration of the predictive schedule $C_{max}^{ref}$. Here, we assume that the decision maker would like that the total completion time of the schedule is not degraded too much by the uncertainties.
- uncertainties are mutually independent, i.e., the consideration and the evolution of a uncertainty do not influence the occurrence or the evolution of another uncertainty.

A schedule can be impacted either by uncertainties on events (UoE) $h \in H$. We distinguish two types of UoP: the set $U^{ex}$ of uncertainties $u^{ex}$ that directly impact the execution

duration of the operations, and the set $U^h$ of UoE $u^h$ that are linked to the duration of the unpredictable event $h \in H$. We define $U^H = \underset{h \in H}{\cup} U^h$.

Moreover, we denote as $NbU^{ex}$ the number of UoP on parameters linked to the execution duration in $U^{ex}$, while $NbH$ denotes the number of UoE and $NbU^h$ the number of UoE linked to one occurring event $U^h$. Then, $NbU$ denotes the total number of uncertainties. The set of uncertainties can thus be defined by $U = H \cup U^{ex} \cup U^H$.

Every execution duration of operation $d_{jkr}$ is impacted by uncertainties. In fact, an execution duration is defined first by a reference execution duration $d_{jkr}^{ref}$, given by deterministic data. Second, a summation of fluctuations linked to the considered uncertainties whether it is UoE or UoP. Each uncertainty $u^{ex}$ will generate a fluctuation $\delta d_{jkr}^{u^{ex}}$ (that can be positive or negative) on the execution duration $d_{jkr}$. The resulting deviation due to this type of uncertainties is thus the sum of fluctuations ($\sum_{u^{ex}=1}^{NbU^{ex}} \delta d_{jkr}^{u^{ex}}$). For considering UoE, each operation can be impacted by one or more unpredictable event (resource failure, supply disruption, etc). Then, every UoE $h \in H$ is linked with:

- An occurrence probability $p\left(h, o_{jk}\right)$ that represents the probability that the event $h$ impacts the operation $o_{jk}$.
- A Boolean variable $H_{jk}^h$ that can be deduced from the probability of occurrence. It is equal to 1 if the operation $o_{jk}$ is impacted by $h$ and 0 there is no impact.
- A duration $d_{jkr}^{ref,h}$ that represents the reference delay generated by the occurrence of $h$.

The reference duration generated by the occurrence of $h$ may also be impacted by one or more uncertainty $u^h$. This uncertainty generates a fluctuation ($\delta d_{jkr}^{u^h}$) on the considered duration. All the fluctuations expressed by a $\delta d$ are random variables that follow known probability distributions assumed to be bounded into intervals such as $[\delta d^-, \delta d^+]$.

With considering all these elements, the execution duration of an operation is defined by the Expression (1).

$$d_{jkr} = d_{jkr}^{ref} + \sum_{i=1}^{NbU^{ex}} \delta d_{jkr}^{u_i^{ex}} + \sum_{h=1}^{NbH} H_{jk}^h \times \left( d_{jkr}^{ref,h} + \sum_{i=1}^{NbU^h} \delta d_{jkr}^{u_i^h} \right) \qquad (1)$$

The total number of uncertainties considered is assessed by (Expression (2)).

$$NbU = NbOp \times \left( NbU^{ex} + NbH + \sum_{h=1}^{NbH} NbU^h \right) \qquad (2)$$

In a flexible context of Industry 4.0, the description of uncertainties for each operation allow a high level of adaptability. In fact, the Expression (2) can be customized for each operation.

### 4.2.2. Pattern for Uncertainty on Event (UoE)

The UoE pattern of (Figure 5a) presents the impact of the occurrence of an umpredictable event on a schedule operation. The purpose of this pattern is to model the fact that an operation $o_{jk}$ is either impacted or not by the event $h \in H$. For that, from an initial state **Idle**, a transition leads to the locality **OpImpacted** with probability $p(h, o_{jk})$ and to the locality **OpNotImpacted** with probability $1 - p(h, o_{jk})$. In order to execute the other models only when all the uncertainties are initialized, a counter $c$ is updated ($c := c + 1$) when the transition is crossed.

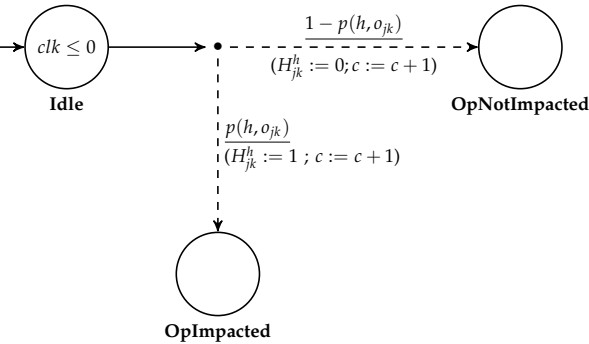

**(a)** UoE pattern.

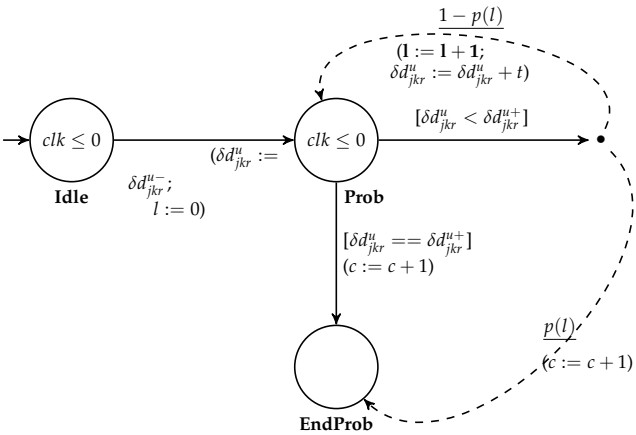

**(b)** UoP pattern.

**Figure 5.** Stochastic patterns of uncertainties.

4.2.3. Pattern for Uncertainty on Parameter (UoP)

The UoP uncertainty pattern presented in (Figure 5b) allows the modelling of the behavior of uncertainties whether it is an uncertainty on execution duration $u^{ex}$ or linked to UoE occurrence $u^h$. The objective of this pattern is to determine the value $\delta d$ in the interval $[\delta d_{jkr}^{u-}, \delta d_{jkr}^{u+}]$ by following a probability distribution. Following the notations given in expression (1), we denote as $\delta d_{jkr}^u$ the duration parameter linked with a discrete random variable $X_d$. The discretization step of the probability distribution is denoted as $t$ and we use an iteration counter denoted as $l$. The philosophy of this pattern is to increase, iteratively and randomly, the value of $\delta d$ in the interval $[\delta d_{jkr}^{u-}, \delta d_{jkr}^{u+}]$.

From the first location of (Figure 5b), the value of $\delta d_{jkr}^u$ and $l$ are initialized. When reaching the location **Prob**, two transitions are enabled depending of the satisfaction of their guards:

1. If the maximum bound of $\delta d$ is reached ($[\delta d_{jkr}^u == \delta d_{jkr}^{u+}]$), the automaton evolves directly to the location **EndProb** and the value of $\delta d_{jkr}^u$ is $\delta d_{jkr}^{u+}$. When crossing this transition, the value of the counter $c$ is also incremented to save the execution of the uncertainty.

2. Otherwise, when $[\delta d_{jkr}^u < \delta d_{jkr}^{u+}]$, $\delta d_{jkr}^u$ is incremented by the discretization step $t$. With crossing this transition, two probabilistic evolutions are possible:

   (a) With the probability $p(l)$, the location **EndProb** is reached which implies that $\delta d$ keeps the value updated during the previous iteration. Moreover, the uncertainty counter is incremented ($c := c + 1$).

(b)　　Otherwise, with the probability $1 - p(l)$ the automaton returns to the location **Prob** with updating the iteration parameter $l$ to $l + 1$. From there, another discretization step is executed such that the value of $\delta d^u_{jkr}$ is updated to $\delta d^u_{jkr} + t$.

In the UoP pattern, the stochastic model is considered in the parameter $p(l)$. To ensure that all uncertainties can be considered regardless of the probability distribution of the stochastic model upcoming from the workshop, we proposed a generic discretization approach (Proposition 1).

**Proposition 1.** *Given a random variable $\delta d^u_{jkr}$ and its known cumulative distribution function $F_X(\delta d^u_{jkr})$, the discrete values of $p(l)$ associated to the fluctuation $\delta d^u_{jkr}$ (Figure 5b) are computed following this equation system (3).*

$$
\begin{cases}
p(0) = \dfrac{F_X(\delta d^{u-}_{jkr}+t) - F_X(\delta d^{u-}_{jkr})}{F_X(\delta d^{u+}_{jkr}+t) - F_X(\delta d^{u-}_{jkr})} \\[3mm]
p(l) = \dfrac{F_X(\delta d^{u-}_{jkr}+(l+1)t) - F_X(\delta d^{u-}_{jkr}+lt)}{F_X(\delta d^{u-}_{jkr}+lt) - F_X(\delta d^{u-}_{jkr}+(l-1)t)} \times \dfrac{p(l-1)}{1-p(l-1)} \qquad for\ l \geq 1
\end{cases}
\tag{3}
$$

This proposition explains how any probability distribution (as long as its cumulative distribution function is known) can be integrated into our models. This contributes to the genericity of our models face to the types of uncertainties.

*4.3. Modelling Schedule Behavior with Uncertainties*

To model the behavior of a schedule, two patterns (operation and resource) are defined and will be instantiated according to the number of operations and resources. When all the uncertainties have been initialized ($c == NbU$), the different fluctuations $\delta d$ generated by the uncertainty patterns are taken into account using the Expression (1) for fixing the duration $d_{jkr}$.

The pattern approach is based on the instantiation principle to model different problems with different sizes. In fact, the instantiation of patterns allows to take into consideration different workshop characteristics without modifying the elements of the patterns, such that we can consider that the models are generic regarding the type of the workshop. The scheduling evolution is represented by the exchange of messages between the operation and resource patterns, denoted as $(Req(r), Comp(o_{jk}))$. The following two paragraphs present the operation and resource patterns.

4.3.1. Operation Pattern

The operation pattern (Figure 6a) models the behavior of one operation $o_{jk}$. The objective of this model is to represent the operation behavior, i.e., (i) checking when the operation can be executed according to the product route and the sequencing on the resources, and (ii) waiting the duration execution on the resource.

After initializing $d_{jkm}$ according to the uncertainties, the workshop constraints are verified through the guard $[Route(o_{jk}) == 1 \ \&\& \ Sequence(o_{jk}) == 1 \ \&\& \ Avail(j) == 1]$ (Precedence on job route if necessary, precedence linked to the sequence of resource and job availability). When the workshop does not consider route constraint (as in the case of Open shop), the condition $Route(o_{jk}) == 1$ is set at true by default. Checking the availability of the job will ensure that there is no overlapping between two or more operations of the same job and ensure the safe execution of the given schedule. The combination of these two constraints guarantees the genericity of the pattern regarding the workshop type.

When the guard is satisfied, the operation $o_{jk}$ synchronizes with its allocated resource $r$ using the request event $(Req(r))$. The availability of the job is then updated and the identifier of the operation is saved ($operation := o_{jk}$). In the **Execution** loca-

tion, the operation is waiting for the end of its execution. The synchronizing event ($Comp(o_{jk})$) allows the operation to cross the transition to its marked location **Completed** with updating the execution status of the operation ($OpComp := 1$) and the availability of its job.

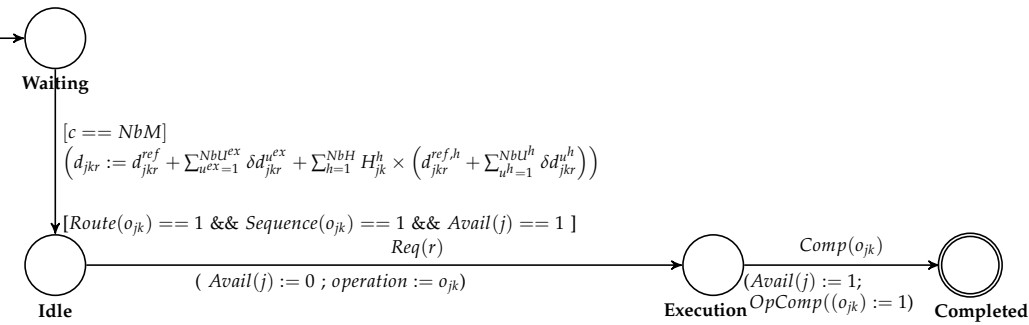

(**a**) Operation pattern.

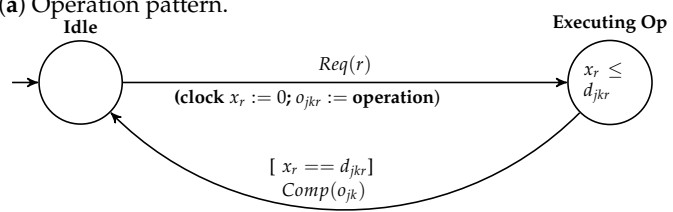

(**b**) Resource pattern.

**Figure 6.** Schedule patterns.

### 4.3.2. Resource Pattern

The resource pattern (Figure 6b) models the behavior of one resource $r$. This model simply represents the state of the resource: available or executing an operation. In the **Idle** location, the resource $r$ is waiting for one operation request ($Req(r)$). When crossing the transition from this location, the local clock is initialized to zero ($x_r := 0$) and the operation identity is saved in the variable $o_{jkr}$. In the location **Executing Op**, the resource starts the execution of the operation. The invariant ($x_r \leq d_{jkr}$) ensures that the duration will not exceed $d_{jkr}$. When reaching the duration value, the guard [$x_r == d_{jkr}$] is then satisfied and the synchronizing event $Comp(o_{jk})$ occurs allowing the resource to go back to the **Idle** location.

### 4.3.3. Evolution of the Modelling Structure

The modelling structure is based on pattern instantiation. In fact all the pattern presented before will be instantiated following the number of operations and resources. The uncertainty patterns will also be instantiated following the set of uncertainties ($U$) and the number of operations. In Figure 7, we use a sequence diagram to illustrate the execution of the structure models. More precisely, the uncertainty models are preliminary executed to compute the duration of each perturbed operation according to the information on the uncertainties. Then, each operation model can be executed according to the previously computed duration and the workshop and schedule constraints. Finally, the resource model is executed for completing the different operations.

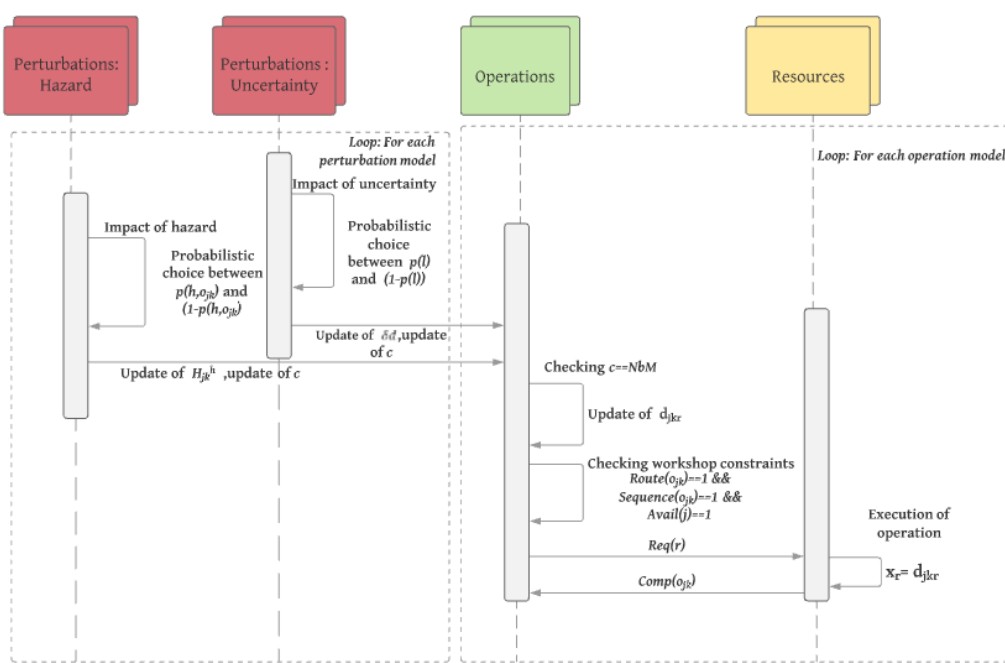

**Figure 7.** Evolution of modelling structure.

## 5. Simulation Based Approach for Robustness Evaluation

### 5.1. Quantifying Robustness Performance

The notion of robustness has different definitions in the literature that converge to the same idea: a robust schedule should maintain or guarantee some performances despite uncertainties and variations generated by the production system or its environment [34].

In this paper, robustness is defined as *the capacity of a schedule to absorb uncertainties without a big decrease in its performance*. The robustness performance therefore seeks to meet the reality of production workshops which is the need to take into consideration the uncertainties in advance when it comes to production control. Therefore, the evaluation of the robustness performance must eventually enable the generation of a robust schedule and the prediction of the scheduling behavior in front of uncertainties. In the continuation of our work, we will focus on this concept of robustness as a performance evaluation to bring elements of response in order to contribute to the challenge of robustness identified in particular by [40] for the implementation of the Industry 4.0. To assess this robustness, a metric has to be defined. In this paper we use the metric defined in [41] that assesses the robustness as a service level. The service level is defined as *"the probability that a criterion is smaller (resp. larger) or equal to a given value"*. In our case, the considered criterion of the scheduling problem is the total completion time, so the service level assessment is the probability that the makespan is smaller (or equal) to a given deadline defined by the decision maker. Formally, this metric is given by the Equation (4) :

$$RL\big(s, U, \tilde{d}\big) = Pr\big(C_{max}(s, U) \leq \tilde{d}\big) \tag{4}$$

Explicitly, the Equation (4) expresses the measure of probability *Pr* that the makespan $C_{max}$ of the schedule *s* subjected to the set of uncertainties *U* is less or equal than a fixed deadline $\tilde{d}$.

### 5.2. Robustness Evaluation by Model-Checking

To assess the robustness of a schedule over the modelling structure given above, we use model checking tools. The main objectives of the model checking approach is first to translate the expression of service level defined in (Expression (4)) in a language that is recognized by STA and second is to find the suitable approach to assess *RL*. In DES domain, model checking is usually used to verify if a model satisfies some properties or not. These

properties can be expressed in several logic languages (CTL, PCTL, etc.). To assess the robustness, we use an extension of model-checking to stochastic DES models. For stochastic DES models, the properties are expressed in PCTL logic (Probabilistic Computation Tree Logic) [42]. This language is a probabilistic extension of CTL (Computation Tree Logic) [43]. This type of logic allows the expressing properties such as *"What is the probability that the model is in the state A, in the precise interval [0,T]?"*. This question can be transcribed in PCTL as in the Expression (5).

$$P = ?[F \leq T \text{ "}A\text{"}] \tag{5}$$

The operator $P = ?$ expresses the probability assessment $Pr$. The operator $F$ means that there exists eventually a path where the automata is in the state "A".

To assess the robustness level of a schedule, the question asked is: *"What is the probability that all paths lead to a global state where all operation models are in the marked location **Completed** in a duration that is less or equal to a given deadline $\tilde{d}$?"*.

Using PCTL logic, the Formula (4) can be expressed as the property:

$$P = ?[F \leq \tilde{d} \text{ "All operations } o_{jk} \text{ are Completed"}] \tag{6}$$

Actually, the Makespan $C_{max}(s, U)$ is given by the value of the global clock ($clk$). The formula $[F \leq \tilde{d} \text{ "All operations } o_{jk} \text{ are Completed"}]$ is a PCTL expression for:

$(C_{max}(s, U) \leq \tilde{d})$. The statistical model checking (SMC) is used in order to check the stochastic models. SMC generates various execution paths and verifies, after each execution, the satisfaction of a property for giving the associated statistical results (in the same way of Monte Carlo simulation). This avoids the combinatory explosion and is then adapted for checking real systems [44].

A confidence interval is an estimated interval having a specified precision $\epsilon$ and a confidence level $\alpha$ such that the robustness level $RL(s, U, \tilde{d})$ belongs to the interval $[RL(s, U, \tilde{d}) - \epsilon; RL(s, U, \tilde{d}) + \epsilon]$ with a chance of $(1 - \alpha)$ [45]. In this paper we use UppAal SMC tool to implement both models and property [46].

## 6. Illustrative Example

In the scenario presented below, we assume that the production environment can undergo several uncertainties. The objective of this application is to illustrate the use of the evaluation process in evaluating schedule robustness on a numerical example. The genericity to the type of workshop and to the uncertainties of the proposed approach, as well as the scaling up will be presented in the next section.

Let us consider the following situation. In a transition to Industry 4.0 transformation, a company has set up a machine learning system to manage the collection and treatment of data [47]. The collected data from one lines are treated and kept in a virtual Data Center and used to contribute into workshop simulation. The treatment of these data allows the setting of knowledge about the uncertainties that may impact production lines.

### 6.1. Problem Description

This line represents a flexible job shop able to produce customized products. The main challenge when scheduling this line is to guarantee a flexible production satisfying a deadline.

To schedule this workshop, the strategy adopted is a proactive one. Where the decision maker generate several schedules with different makespans. The historical data of this workshop shows that machine breakdown and uncertainty on execution duration have an irreversible impact on the total duration of schedule and should be considered in workshop scheduling.

The characteristics of the problem are as follows:

- Workshop size : $NbJ = 8, NbOp = 35, NbR = 7$,

- First UoE $h_1$ : machine breakdown. We consider that during the scheduling horizon at least one operation is impacted by the failing of a resource. This is translated by an occurrence probability of $p(h, o_{jk}) = 3\%, \forall(j, k)$ ($35 \times 3\% \approx 1$).
- Second UoP $u_1^h$: Uncertainty on maintenance duration. In fact when a machine breakdown occur in the workshop, usually the mean time of maintenance is considered. In reality, this duration can variate following the type of failure and the availability of human resources. To consider this variation, the stochastic model constructed is based on an exponential distribution.
- Third UoP $u_1^{ex}$: Uncertainty on execution duration. When the schedule is executed on this line, the effective execution duration of operations is deviated from the reference durations defined. With the historical data recovered, the uncertainty on execution durations are to be considered in an exponential probability distribution.

### 6.2. Illustration Results and Discussion

In UppAal SMC tool, the two parameters to set are named $\alpha$ and $\epsilon$. The parameter $\alpha$ defines the risk for the calculated probability to be outside the confidence interval.

To have the most precise probability value, we set the parameters $\alpha$ and $\epsilon$ to respectively 2% and 1% for the implementation of the evaluation process. The scheduler has generated three schedules to be implemented in the workshop ($S = \{s_1, s_2, s_3\}$) with close reference makespans $C_{max}^{ref}(s)$ but different sequencing on resources. The set of uncertainties considered is $U = u_1^{ex}, h_1, u_1^{h_1}$ and the deadline to respect is defined as a spread on reference makespan $\tilde{d} = 110\% C_{max}^{ref}$.

The evaluation process is used to assess the service level of each schedule in front of expected uncertainties (uncertainty on execution duration, machine breakdown and uncertainty on maintenance duration). The results of this application are given in (Table 2)

**Table 2.** Robustness evaluation results.

| Schedule | $C_{max}^{ref}(TU)$ | $\tilde{d}(TU)$ | CI of $RL_i$ | $RL_i$ |
|:---:|:---:|:---:|:---:|:---:|
| $s_1$ | 38 | 42 | $[0.69, 0.71]$ | 70% |
| $s_2$ | 39 | 43 | $[0.75, 0.77]$ | 76% |
| $s_3$ | 42 | 46 | $[0.71, 0.73]$ | 72% |

With the resulted information, the decision maker can interpret the results and choose a schedule to be executed in the workshop. For instance, it can be concluded that the solution $s_2$ is the most robust (the highest service level $RL_2 = 76\%$). Moreover, this solution represents a good compromise between robustness and time performance (Makespan).

## 7. Assessment of Process Performances

To implement the evaluation process in the context of Industry 4.0, its ability to reach some performances must be checked. We consider that this process must satisfy the challenges identified in the research questions. Two main performance criteria are identified: first the genericity of the evaluation process and second its sensitivity.

### 7.1. Process Genericity

One of the initial arguments for the definition and the implementation of the proposed process was based on the observation that the existing methods are often dedicated to a specific scheduling problem (type of workshop considered, uncertainties considered). Thus, the objective here is to evaluate to what degree the proposed approach is generic. Two questions are addressed:

- Is the approach generic to the type of production workshop considered?
  *Being used with different workshops (Flexible job shop, Flow shop and parallel machines). Indeed, the instantiation of the operation pattern allows to consider the different workshop*

*constraints. With this characteristic, the evaluation process can be used in the flexible context of Industry 4.0 where the workshop configuration is intended to be constantly moving.*

- Is the approach generic with to the considered disturbances?
  *Being used with several uncertainties. The fact here is that as long as the uncertainty is a UoE or an uncertainty and its impact can be translated into fluctuations on the runtime, then it can be integrated in the evaluation process.*

### 7.2. Process Sensitivity and Scalability

For the purpose of implementation, it is also important to analyze the performance of our approach in terms of resolution. The following performance concerns the computation time. In order to analyze the performances of the process, two parameters are studied: input parameters and size of the workshop. First, the purpose is to test process performances to the variation of inputs given by the decision maker. The input parameters treated are the initial given schedule, the uncertainties and the deadline. The second parameter analyzed is the size of the workshop. Here, the sensitivity to the variation of workshop's size is treated. In the same time, the scalability of the process is demonstrated.

To conduct a successful experiment, we defined a method based on experimental design. The experimental design method enables the organization of the experimentation phase while optimizing the number of necessary experiments. The purpose of a design of experiments is to establish a link between a response $Y$ and factors $X_i$ ($Y = F(X_i)$). In our case, we are interested in the response computation time and the factors are the different input parameters of the evaluation process.

The experimental design proposed to analyze the sensitivity of the process is based on several factors having three levels (Table 3). For each treated factor, three levels are defined in order to construct a varied experimental panel. Indeed, this panel treats different input schedules ($s_1, s_2, s_3$) and different uncertainties parameters, the variation $\delta$, the probability distribution $p(l)$ and the occurrence probability of UoEs $p(h, o_{jk})$. The last factor is the deadline defined $\tilde{d}$.

**Table 3.** Experimental design parameters for process sensitivity analysis.

| Factor Level | $s$ | $\delta d_{jkr}^{u^{ex}}$ | $p(l)$ | $\delta d_{jkr}^{u^h}$ | $p(h, o_{jk})$ | $\tilde{d}$ |
|---|---|---|---|---|---|---|
| 1 | $s_1$ | $+/-$ $10\% d_{jkr}^{ref}$ | $expon$ | $+/-$ $10\% d_{jkr}^{ref,h}$ | 1% | $110\% C_{max}^{ref}$ |
| 2 | $s_2$ | $+/-$ $25\% d_{jkr}^{ref}$ | $unif$ | $+/-$ $25\% d_{jkr}^{ref,h}$ | 3% | $120\% C_{max}^{ref}$ |
| 3 | $s_3$ | $+/-$ $40\% d_{jkr}^{ref}$ | $norm$ | $+/-$ $40\% d_{jkr}^{ref,h}$ | 5% | $130\% C_{max}^{ref}$ |

The second experimental design is designated to treat the scalability of the process in front of problem size (Table 4). The considered factors are the workshop size parameters, the number of jobs $NbJ$, the number of operations per job $NbOp^j$ and the number of resources $NbR$. From the Table 4, an experimental panel with different workshop sizes are treated.

**Table 4.** Experimental design parameters for process scalability analysis.

| Factor Level | $NbJ$ | $NbOp^j$ | $NbR$ |
|---|---|---|---|
| 1 | 8 | $1([3;7])$ | 7 |
| 2 | 16 | $2([6;14])$ | 14 |
| 3 | 32 | $3([12,28])$ | 28 |

### 7.3. Sensitivity and Scalability Discussion

For the sensitivity of the evaluation process to the input parameters, the following observations are made and summarized in Table 5. The analysis of the factors related to the uncertainty parameters allows us to conclude that for the uncertainties, the variability of the duration does not have a statistically significant effect on the performance of the approach, but the probability distribution necessary for the calculation of $p(l)$ does have an effect on the performance of the process. For the UoE, the probability $p(h, o_{jk})$ of occurrence has a considerable effect on the performance of the process. Indeed, as the value of this probability increases, the computation time also increases. This deterioration is explained by the increase of the number of operations impacted by the UoE. Indeed, by increasing $p(h, o_{jk})$, the number of operations that can be affected by the UoE increases. When the deadline increases, the computation time decreases.

**Table 5.** Factors impact on time performance.

| Factor | $s$ | $\delta d_{jkr}^{ref}$ | $p(l)$ | $\delta d_{jkr}^{ref,h}$ | $p(h, o_{jk})$ | $\tilde{d}$ |
|--------|-----|------------------------|--------|--------------------------|----------------|-------------|
| Effect | No | Yes | Yes | Yes | Yes | Yes |

For the scalability analysis, the three factors of workshop size have an effect on the performance of the evaluation process. The main effects graphs allow to see that the three factors have a visible impact on the computation time (Figure 8b). In fact, the evolution of the computation time increases drastically with the size of the workshop.

In order to understand the impact of the size of the workshop on the performances of the approach, we define a new parameter: the potential charge of the workshop. This is defined as the ratio between the number of operations to be executed in the workshop ($NbOp$) and the number of available resources ($NbR$) (Figure 8a). The ratio $NbOp/NbR$ impacts significantly the computation time.

From the sensitivity analysis, we could deduce that indeed the input parameters of the problem have an impact on the resolution time of the process. A problem with multiple uncertainties and with complex parameters will probably increase the resolution time of the process. This sensitivity is considered as an acceptable price of genericity. The fact that the proposed process can treat different problems with different workshop configuration and different scenarii of uncertainties without any modelling effort. The process is based on an instantiation approach, which makes it easier to adapt it to several instances. When proposing an optimal, dedicated model for one problem, this sensitivity is less important.

For the scalability analysis, the evaluation process has been applied on different workshop sizes up to 560 operations to be executed on 28 resources. This meets the reality of production workshops in biggest companies and allow to confirm the ability to use the process in a real industrial case. It is true that the resolution time increases exponentially with the size of the workshop but from all the instances treated in the experimental panel, we could propose a solution in an average time of 50 min which is realistic. For small instances, the resolution time do not exceed 3 s. With the obtained results, the evaluation process implementation can be intended.

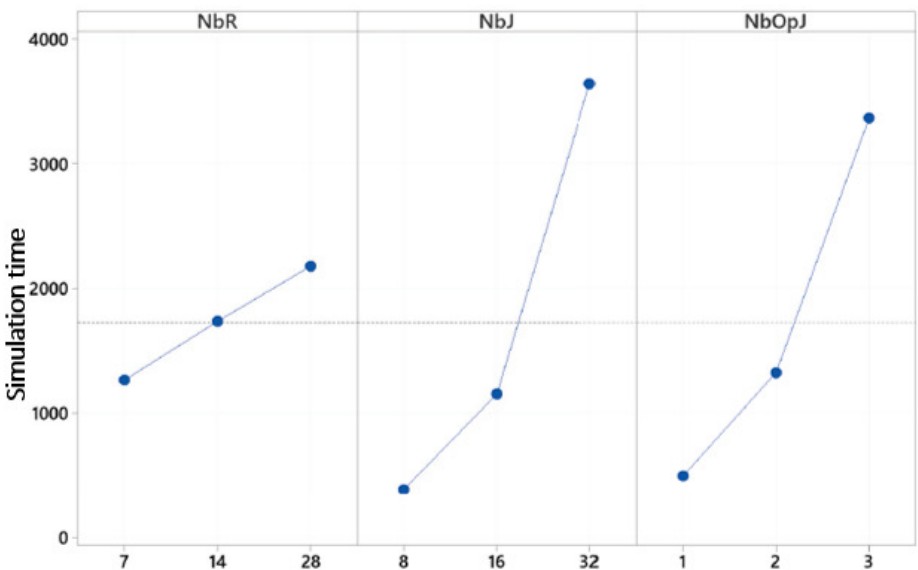

(**a**) Effect of the resource charge on resolution Time.

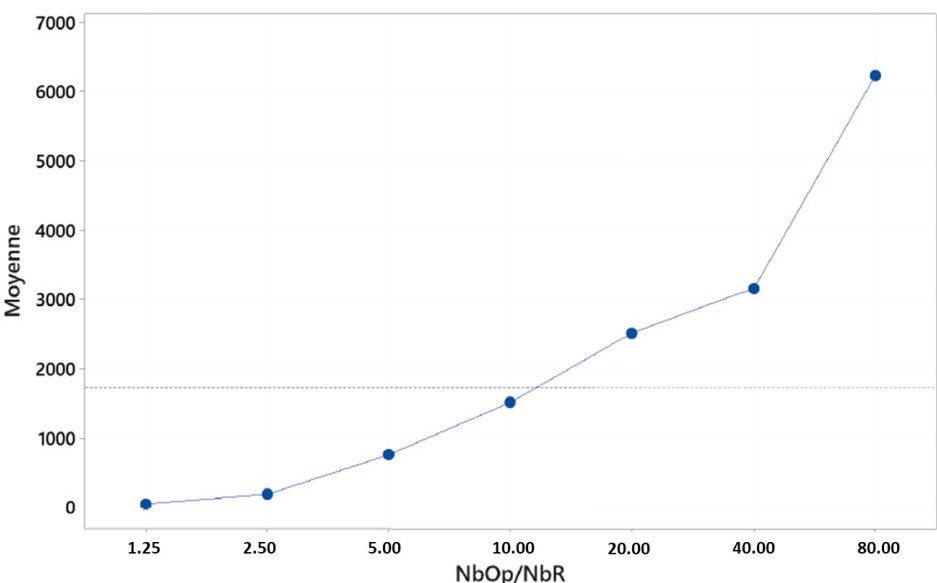

(**b**) Effect of workshop size on resolution Time.

**Figure 8.** Scalability analysis of the evaluation process.

## 8. Conclusions and Perspectives

In this paper, a generic and reusable approach for evaluating the robustness of schedules under uncertainties is proposed. This approach contributes to the issue of robust scheduling in the context of Industry 4.0. Existing approaches in the literature are often dedicated to the type of workshops and uncertainties and dependent to the evaluated performance. These assumptions are difficult to hold in the context of Industry 4.0, where workshops must be flexible and agile. It has therefore been shown that it is important to have a generic approach that can be adapted to any type of workshop and uncertainties. To answer this research question, the robustness performance is considered. In fact, this paper proposes a formalization of robust scheduling problems based on the service level. To evaluate this service level, the proposed modelling approach is based on stochastic timed automata. This approach is both modular and generic, since it is based on modules representing the behavior of the schedule ( operation patterns, resource patterns) and the

behaviour of uncertainties ( UoP patterns, UoE patterns). The assessment method of service level is based on formal verification using statistical model checking. The process feasibility is illustrated on a flexible job shop with dealing with machined failure and uncertainties on maintenance durations and on execution durations. This illustration highlighted the usability of process results in decision-making support. The analysis of the process performances demonstrates the genericity of the process to the type of workshop considered. The adaptability of the process to different types of uncertainties is also highlighted. Moreover, the analysis of process sensitivity and scalability has prooven the usability of the process into real world cases.

The data of the production workshop used as input of the evaluation process is considered as granted. A first perspective would be to integrate a data-mining method in order to learn from the real behavior of the workshop. With this perspective comes the implementation of the robustness evaluation process in a simulation method in order to support the decision making and increase the reactivity of the decisions to the upcoming changes in the workshop. Another perspective would be to create a hybrid approach based on Operational Research methods to optimize the robustness of the schedule. In fact we have seen that several scheduling techniques existing in the literature needs at some point to proceed to an evaluation of schedule performance. One interesting perspective would be to study the possible combination of these techniques with the proposed evaluation process.

**Author Contributions:** All the authors contributed equally to the conceptualization and methodology; investigation, validation, formal analysis have been conducted mainly by S.H.; the writing—original draft preparation has been performed equally by S.H., P.M. and A.A. whereas J.-F.P. verified the document; visualization has been done equally by S.H., P.M. and A.A. All authors have read and agreed to the published version of the manuscript.

**Funding:** This research received no external funding.

**Data Availability Statement:** The data presented in this study are available on request from the corresponding author.

**Conflicts of Interest:** The authors declare no conflict of interest.

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
