# Peer review of "Robustness Evaluation Process for Scheduling under Uncertainties"

_processes, doi:10.3390/pr11020371_

Round 1
Reviewer 1 Report
Manuscript ID processes – 2068574
Title: "Robustness evaluation process for scheduling under uncertainties".
Undoubtedly, the presented manuscript is relevant from a scientific and practical point of view, especially in the context of the industry 4.0. This study opens up defined prospects in this field of knowledge. Manuscript entitled "Robustness evaluation process for scheduling under uncertainties" of interest for a highly ranked journal like "Processes".
This manuscript includes the next harmonious structure:
* Introduction (p. 1 – 2);
* Literature review (p. 2 – 5);
* Process for robustness evaluation of scheduling under uncertainties (p. 5 – 15);
* Numerical example (p. 15 – 16);
* Assessment of process performances (p. 16 – 18);
* Conclusion and perspectives (p. 18 – 19).
Weakness and methodological inaccuracies are not detected. The figures and tables are appropriate; they reflect the complex results of this study. References list is adequate and includes 45 titles.
In my opinion, the value of this article is significant.
Author Response
We would like to thank the reviewer fro appreciating our paper. We made some modifications in the document such that we hope the presentation of the results has been improved :
- restructuration of the document
- explaining deeper the link between contributions and research questions
- introducing the goal of the experiments (sections dedicated to the numerical example and the assessment of performances).
Reviewer 2 Report
Dear Authors,
The topic you have chosen is topical because companies are under pressure to serve their customers with the right quality and at the right time.
Abstract
Very general. Does not give a clear answer why Stochastic Discrete Event Systems models are proposed. What is the result? How have the results been validated?
Introduction
Correct. In my opinion the section from line 46 to line 51 is not important.
"The remaining of the......... further perspectives are given."
Literature review
Please expand the references. There is a large literature on the modelling of scheduling tasks. E.g.
Kocsi, Balázs ; Matonya, Michael Maiko ; Pusztai, László Péter ; Budai, István
Real-Time Decision-Support System for High-Mix Low-Volume Production Scheduling in Industry 4.0
PROCESSES 8 : 8 pp. 1-26. Paper: 912 , 26 p. (2020)
There are several places where the topic could be addressed in more detail using fuzzy methods
e.g. fuzzy description - line 120
Pusztai, László ; Kocsi, Balázs ; Budai, István
Making engineering projects more thoughtful with the use of fuzzy value-based project planning
POLLACK PERIODICA: AN INTERNATIONAL JOURNAL FOR ENGINEERING AND INFORMATION SCIENCES 14 : 1 pp. 25-34. , 10 p. (2019)
line 130
It is not entirely clear why Schedule robustness is a performance indicator. There are several indicators that can be used.
Pusztai, László Péter; Nagy, Lajos ; Budai, István
Selection of Production Reliability Indicators for Project Simulation Model
APPLIED SCIENCES-BASEL 12 : 10 Paper: 5012 , 15 p. (2022)
chapter 2.2. Discussion and solution direction
The research questions overlap. If several questions are asked, which answer answers which question?
Chapter 3 Process for robustness evaluation of scheduling under uncertainties
Correct.
Chapter 4. Numerical example
The shortcoming of this chapter is that it uses only one Workshop size parameter. It would be expected to use at least two types of Workshop size.
Other observation
Fig 2 , Fig 7 , Fig 8 not readable size,
Please correct your manuscript based on the above
Round 2
Reviewer 2 Report
Dear Authors,
Thank you for the corrections in your manuscript. I accept it.